Temporally-aware algorithms for the classification of anuran sounds

Luque Amalia 1 amalialuque@us.es
http://orcid.org/0000-0002-6456-7036 Romero-Lemos Javier 1
http://orcid.org/0000-0001-9474-3929 Carrasco Alejandro 2
Gonzalez-Abril Luis 3
1 Departamento de Ingeniería del Diseño, Universidad de Sevilla , Sevilla , Spain
2 Departamento de Tecnología Electrónica, Universidad de Sevilla , Sevilla , Spain
3 Departamento de Economía Aplicada I, Universidad de Sevilla , Sevilla , Spain
Brook Barry
Electronic publication date: 2018 May 4
Publication date: 2018
Volume: 6
Electronic Location ID: e4732
Received 2017 Dec 19; Accepted 2018 Apr 18
Copyright: © 2018 Luque et al.
Copyright year: 2018
Copyright holder: Luque et al.
License: This is an open access article distributed under the terms of the Creative Commons Attribution License, which permits unrestricted use, distribution, reproduction and adaptation in any medium and for any purpose provided that it is properly attributed. For attribution, the original author(s), title, publication source (PeerJ) and either DOI or URL of the article must be cited.
License URL: https://creativecommons.org/licenses/by/4.0/

Keywords: Global warming, Sound classification, Data mining, Feature extraction, Machine learning, Habitat monitoring

Funding: Consejería de Innovación, Ciencia y Empresa, Junta de Andalucía, Spain This work has been supported by the Consejería de Innovación, Ciencia y Empresa, Junta de Andalucía, Spain, through the excellence eSAPIENS (reference number TIC-5705). The funders had no role in study design, data collection and analysis, decision to publish, or preparation of the manuscript.

==============================
Several authors have shown that the sounds of anurans can be used as an indicator of climate change. Hence, the recording, storage and further processing of a huge number of anuran sounds, distributed over time and space, are required in order to obtain this indicator. Furthermore, it is desirable to have algorithms and tools for the automatic classification of the different classes of sounds. In this paper, six classification methods are proposed, all based on the data-mining domain, which strive to take advantage of the temporal character of the sounds. The definition and comparison of these classification methods is undertaken using several approaches. The main conclusions of this paper are that: (i) the sliding window method attained the best results in the experiments presented, and even outperformed the hidden Markov models usually employed in similar applications; (ii) noteworthy overall classification performance has been obtained, which is an especially striking result considering that the sounds analysed were affected by a highly noisy background; (iii) the instance selection for the determination of the sounds in the training dataset offers better results than cross-validation techniques; and (iv) the temporally-aware classifiers have revealed that they can obtain better performance than their non-temporally-aware counterparts.

Introduction

Sound classification has become a major issue in numerous scientific and technical applications. Many techniques have been proposed to obtain the desired sound labelling: some for general purpose (Hinton et al., 2012) and others for specific applications (Cowling & Sitte, 2003).

Although sounds are inherently represented by a time series of acoustic data, it is common to focus on small fragments of audio signals and attempt to classify them without considering the preceding or subsequent sound sections. For this reason, non-temporally-aware (NTA) methods are also frequently applied (Tzanetakis & Cook, 2002; Wang et al., 2006).

In order to clarify the temporal character of the sound in this paper, our interest lies in the evolution of its low-level short-duration frames and not to the sequence of acoustic units commonly used in bioacoustics (Kershenbaum et al., 2016).

The main aim of this paper is to analyse and compare temporally-aware and NTA classifiers and to show that the consideration of temporal information clearly improves classification performance.

Let us indicate that the study presented here could eventually be applied to the study of global warming, since the sounds produced by certain animal species have been revealed as a strong indicator of temperature changes and, therefore, of the existence of climate change. Of particular interest are the results provided by anuran-sound analysis (Márquez & Bosch, 1995), and hence these kinds of sounds are analysed in this paper.

As a widely distributed taxonomic group, anurans are considered excellent indicators of biodiversity. However, frog populations have been experiencing dramatic declines over the past decade due to habitat loss, climate change, and invasive species (Xie et al., 2017). Therefore, long-term monitoring of frog populations is becoming increasingly important in the optimization of conservation policy.

It is worth noting that the sound production mechanism in ectotherms is strongly influenced by the ambient temperature (Llusia et al., 2013). Hence, the temperature may significantly affect the patterns of calling songs by modifying the beginning, duration, and intensity of calling episodes and, consequently, the anuran reproductive activity. The presence or absence of certain anuran calls in a certain region, and their evolution over time, can therefore be used as an indicator of climate change.

The first step in biological species identification involves the recording of different sounds in their natural environment, where different devices can be used. Processing of the recorded sounds can be performed either locally in real time (Aide et al., 2013), or in a remote centre requiring, in this case, a suitable communication system, usually a wireless sensor network, which generally requires information-compressing technologies (Diaz et al., 2012).

In previous work (Luque et al., 2016), a NTA method for sound classification has been proposed. According to this procedure, the sound is split up into frames. Every frame is subsequently featured using 18 parameters (also called features). The frame features are then compared to certain frame patterns belonging to known sounds, thereby assigning a class label to each frame. Finally the sound is classified by frame voting, for which up to nine different algorithms have been proposed (Luque et al., 2018; Romero, Luque & Carrasco, 2016). For the determination of the sounds which should be included in the training dataset, instance selection and cross-validation techniques are considered and compared.

However, sounds are inherently made up of a time series of acoustic data. Therefore, if the temporal information of the frame is added to the classification process, then better classification results should be expected. It must be borne in mind that the goal of classification is to recognize species and, more precisely, their different vocalizations.

The paper is organized as follows: section “Materials and Methods” describes the anuran dataset and presents the methodology employed to compare classifiers, by depicting its general schema, the six (three frame-based, three segment-based) approaches to temporally-aware classification, the classification algorithms considered, and the performance metrics employed. The application for the classification of a set of actual anuran sounds is presented in section “Results,” where the results of the six approaches are compared with each other and also with NTA classifiers.

Materials and Methods

Sound dataset

For testing purposes, actual anuran sounds provided by the National Natural History Museum (Fonozoo.com, 2017) have been employed. The sounds correspond to two species, the Epidalea calamita (natterjack toad) and Alytes obstetricans (common midwife toad), with a total of 868 recordings containing four classes of sounds: E. calamita; mating call (369 records).

E. calamita; release call (63 records).

A. obstetricans; mating call (419 records).

A. obstetricans; distress call (17 records).

A total of 4,343 s of recording have been analysed, with an average duration of 5 s. A common feature of all the recordings is that they have been taken in their natural habitat, with very significant surrounding noise (wind, water, rain, traffic, voices, etc.), which posed an additional challenge in the classification process.

To perform a supervised classification, certain sounds have to be selected as patterns (to be used in the training phase) and others are employed for validation and testing purposes. A common practice is to split the dataset into several disjoint subsets and apply a cross-validation technique (four folds have been used in the paper). However, use of these noisy recordings as patterns may lead to a decrease in the classification performance. Hence, several other approaches arise as an alternative to cross-validation. In our case, recordings with relatively low background noise, which were carefully selected by biologists and sound engineers, have been used as patterns.

This approach, usually called instance or example selection, is recommended in order to increase the rate of learning by focusing attention on informative examples (Blum & Langley, 1997; Raman & Ioerger, 2003; Olvera-López et al., 2010; Borovicka et al., 2012). In order to determine the frame patterns, the experts listen to the recording of the anuran calls and simultaneously consider the spectrogram and the set of MPEG-7 features, and label each frame that they consider may belong to any of the possible classes.

The parameters for every classifier are determined by exclusively using the pattern records (training dataset). The remaining elements in the dataset are then divided into two approximately equal subsets used for validation and testing. The validation dataset is employed to determine the hyper-parameters of the classifiers, such as the number r of relevant features, the number w of frames to be considered in sliding window (SW), recurrent sliding window (RSW) and hidden Markov model (HMM)–SW, and the number T of recurrent inputs in recurrent neural networks (RNNs). On the other hand, the testing dataset, which includes none of the patterns nor validation sounds, is employed for the evaluation of the performance of every algorithm. Table 1 summarizes the sound and pattern dataset.

Table 1 Sound and pattern datasets.

Sound class	Sound records	Pattern records	
Number	Seconds	Number	Seconds (pattern section)	Seconds (total record)	
Ep. cal. mating call	369 (43%)	1,853	4	13.89	20.39	
Ep. cal. release call	63 (7%)	311	3	0.99	14.56	
Al. ob. mating call	419 (48%)	2,096	4	1.09	19.72	
Al. ob. distress call	17 (2%)	83	2	3.30	9.80	
Silence/noise	–	–	–	45.20	–	
Total	868	4,343	13	64.47	64.47	

General description of the classification methodology

The general schema of the proposed procedure, depicted in Fig. 1, is based on the following steps: The sound is split up into 10 ms frames. This is the frame length recommended by MPEG-7 since it is the approximate time period for the opening and closing of the anuran vocal cords.

Every frame uses D MPEG-7 features: a vector x in ℝD (ISO, 2001). The series of S frame vectors [x1, x2 … xS] makes up the X matrix of dimension M × D, which features the full sound. Feature extraction using MPEG-7 features has been chosen because very good results are shown in their description of sound frames for classification purposes, and these features appear as serious competitors to mel-frequency cepstral coefficients (MFCC) features, which are widely used in many applications (Herrera-Boyer, Peeters & Dubnov, 2003). MPEG-7 and MFCC features show similar classification performance and although MPEG-7 extraction could require more computational effort, it enjoys several advantages: it is semantically richer (in the sense that it is easier to intuitively grasp its meaning); it is fully standardized for general-purpose applications; and it presents better performance when a reduction in the number of features is required.

Temporal information is considered by using one of two approaches: 3.1 Frame-based approach: for every frame, a vector y of C additional features is constructed (Liu & Motoda, 1998) by applying a function f to the matrix X of the MPEG-7 original features and hence y = f(X). Therefore, every frame is featured using D + C features, that is, a vector z = x ∪ y in ℝD+C. Certain forms of the function f, for instance, are comprised of statistical measures of X, or the concatenation of the vectors corresponding to stacked frames.

3.2 Segment-based approach: every series of N sound frames is represented by a model using its N × D features.

Every sound fragment, either in terms of a frame or segment, is classified by using one of two approaches: 4.1 The frame features are compared to frame patterns belonging to known species, thereby assigning a class label to each frame. By means of the feature extraction and construction procedures previously described, each sound frame is characterized by D + C features or, equivalently, by a point in an ℝD+C space defined by its coordinate vector z. N pattern frames are also available where the i-th pattern is also represented by a point in the ℝD+C space with a coordinate vector πi. Each frame is labelled as belonging to a certain class θ out of a total of M classes. The set of pattern frames can be viewed as a cloud of points in ℝD+C and can be identified by a matrix Π = [π1, π2, …, πN]′ containing the coordinate vectors of the N points. The subset of points in Π belonging to the class θ is denoted by its matrix Πθ. NTA classifiers perform a certain type of comparison between the frame to be classified (represented by its vector y) and the pattern frames (represented by its matrix Π). This comparison is carried out in the space of the ℝD+C features and its result is called a supervised classification. A wide and representative set of non-sequential supervised classifiers is considered and these are described below.

4.2 The segment models are compared to segment patterns belonging to known species, and a class label is thereby assigned to each segment.

Finally the sound is classified by means of frame or segment voting.

Figure 1 General schema for the classification procedure.

MPEG-7 feature extraction and selection

The task of extracting MPEG-7 features from every sound frame is accomplished by using three different processes: spectrogram analysis, linear prediction coding, and harmonicity analysis. Hence, D = 18 features are obtained by following Luque et al. (2016), which is summarized in Table 2.

Table 2 MPEG-7 features and the processes for their extraction.

	Feature description	Extracting process	
1	Total power	Spectrogram analysis	
2	Relevant power(power in a certain frequency band)	
3	Power centroid	
4	Spectral dispersion	
5	Spectrum flatness	
6,7,8	Frequency of the formants (×3) (The first three formants are considered)	Linear prediction coding	
9,10,11	Bandwidth of the formants (×3) (The first three formants are considered)	
12	Pitch	
13	Harmonic centroid	
14	Harmonic spectral deviation	
15	Harmonic spectral spread	
16	Harmonic spectral variation	
17	Harmonicity ratio	Harmonicity analysis	
18	Upper limit of harmonicity	

As shown below, the consideration of temporal information associated to the frames usually leads us to significantly increase the number of features required. In order to cope with this drawback, a reduction in the number of the 18 original MPEG-7 features is proposed by considering the r most significant features of each frame (leading to a vector in ℝr). Feature selection procedures are employed to determine the relevance-ordered set of features and its optimal size (Guyon et al., 2006).

The feature selection technique used in the paper is based on the Jensen–Shannon divergence (Lin, 1991). It obtains the separability of the sound classes for every feature by applying the following procedure: Consider the set of the N pattern frames represented by the matrix Π = [π1, π2, …, πN]′. Focus on the subset of elements in Π belonging to the k-th class θk, which is denoted by its matrix Πk and, specifically, in the i-th pattern frame πi ∈ Πk. The vector πi contains D elements, one for each feature. The j-th feature corresponding to the i-th pattern frame is denoted as πji | πi ∈ Πk. Let us denote φjk as the set of values of the j-th feature in every frame belonging to the k-th class (θk): φjk = {πji} ∀i | πi ∈ Πk.

Estimate the probability density function (pdf) fjk of the values in ϕjk, that is, of the j-th feature values for those pattern frames belonging to the k-th class.

For the j-th feature and every pair of classes u and v, an indication is obtained of how separate the corresponding fju and fjv pdfs are. For this purpose, the Jensen–Shannon divergence is used, which is given by (1) DJS(fju,fjv)=12∫∞−∞fjulog22fjufju+fjvdx+12∫∞−∞fjvlog22fjvfju+fjvdx.

Every value of the Jensen–Shannon divergence is transformed in the corresponding distance, which is given by (2) dJS(fju,fjv)≡DJS(fju,fjv).

For the j-th feature, the separability index Ψj is derived, in accordance with (3) Ψj≡∏u=1A−1∏v=i+1AdJS(fju,fjv)B,

where A is the total number of classes and B is the number of pairs of classes u and v, which is given by (4) B=A(A−1)2.

The separability index Ψj for the j-th feature is an indicator of how separate the pdfs are corresponding to each class. The more separate the pdfs are, the more useful (or relevant, or significant) that feature is for classification. Hence, the value of Ψj is used as an indicator of the relevance of the j-th feature.

For comparison purposes, two NTA methods are also considered: NTA classification based on 18 MPEG-7 features (NS-18).

NTA classification based on the r most relevant MPEG-7 features (NS-r).

Feature construction

In order to consider the temporal behaviour of a sound, the frames should not be considered one by one, but the preceding and subsequent frames should also be taken into account, that is, their ordered succession should be considered. Several methods have been proposed in the literature to include this temporality (Dietterich, 2002; Esling & Agon, 2012). A number of these methods can be considered frame-based, that is, they still classify frames but now the frames are featured with additional information on the temporal context. Alternatively, other approaches are defined as segment-based as they do not classify isolated frames but instead classify a series of frames (a segment). First, three frame-based approaches are described: Construction of local interquartile range (LIQR) features (Schaidnagel, Connolly & Laux, 2014). The general idea for this feature construction technique is to use the time axis to construct new temporally-aware features. These techniques are commonly based on the values of the features of the frame without considering their order, which is usually called a bag of features. Average values or other related statistics are usually employed.

In the case of the anuran calls to be classified, the typical croaking of a frog is found, while other calls are similar to the sound of a whistle. The croaking sound is produced by repeatedly opening and closing the vocal cords (roughly every 10 ms, equal to the frame length) leading to a series of frames featured with widely spread values (Fay, 2012). On the other hand, the whistle-like sounds are produced by a continuous air flow showing a narrow spread in feature values. For the incorporation of this information into the classification process, a new set of features is therefore constructed that considers the spread of the extracted feature values and not their average. Furthermore, in order to avoid the influence of outliers, the interquartile range (IQR) is selected instead of the standard deviation.

In the implementation used, first for every frame, a ‘window’ centred on that frame is considered, using the closest neighbouring frames. For every original feature, a new derived feature is constructed. To this end, the values of the original feature for every frame in the window are considered. The IQR of these values is computed, and this value is considered the new derived feature. In this way, the number of constructed features is C = D, and hence up to 2 × D features (a vector in ℝ2×D) are now identifying a frame, where C of these features include temporal information. In this approach, a window size of 10 frames (100 ms) has been used.

SW (Aggarwal, 2007). In this technique, also known as frame stacking or shingling, a short window with w frames, centred on each frame, is considered. An odd-numbered value is usually chosen for the window size, that is, w = 2d + 1, where d is an integer. The class θi for the i-th frame is obtained using a classifying function fc, as follows: (5) θi=fC(xi−d,⋯,xi−1,xi,xi+1,⋯xi+d),

where xj ∈ ℝD represents the feature vector for the j-th frame. The D features describing each frame are now those corresponding to all the frames under the window. Therefore, each frame is featured using w × D features (a vector in ℝw×D). In this approach, the number of features describing each frame can significantly increase, thereby inflicting a major impact on the computing resources required in the classification process. For this reason, only the r most relevant features have been used by applying the aforementioned feature selection techniques.

RSW (Joshi & Dietterich, 2003). This is a method similar to the SW procedure above, except that the classifier now considers not only the features of the frame under the window, but also their previous classification results. Thus, the class θi for the i-th frame is obtained as follows: (6) θi=fC(θi−d,⋯,θi−1,xi−d,⋯,xi,⋯xi+d).

NTA classifiers

Every frame-based approach (and also the segment-based autoregressive integrated moving average (ARIMA) model, described below) relies on an underlying NTA classifier. A broad and representative selection of these classifiers has been used throughout this paper: minimum distance (MinDis) (Wacker & Landgrebe, 1971); maximum likelihood (MaxLik) (Le Cam, 1990); decision tree (DecTr) (Rokach & Maimon, 2008); k-nearest neighbours (kNN) (Cover & Hart, 1967); support vector machine (SVM) (Vapnik, 1998); logistic regression (LogReg) (Dobson & Barnett, 2008); neural network (Neur) (Du & Swamy, 2013); discriminant function (Discr) (Härdle & Simar, 2012); and Bayesian classifier (Bayes) (Hastie, Tibshirani & Friedman, 2005).

All these classifiers have been prototyped using MATLAB. The minimum distance classifier in its training phase obtains the mean value μjk for the j-th feature belonging to the k-th class. In the test phase for the i-th frame, the distance dik between the frame features and the mean value of the k-th class θk is obtained in accordance with the expression (7) dik=∑j=1D(xji−μjk)2,

where xji is the value of the j-th feature corresponding to the i-th frame. The class assigned to the frame is that with the minimum distance.

The maximum likelihood classifier is used under a Gaussian probability distribution with full covariance. The neural network classifier is based on a feed-forward neural network with a 10-neuron hidden layer and a one-neuron output layer. The remaining methods and classifiers have been coded based on built-in MATLAB functions using their default parameters, which are reflected in Table 3.

Table 3 MATLAB functions supporting the various classifiers.

Classif.	Training function	Test function	Additional function	
MinDis	–	–		
MaxLik	fitgmdist	mvnpdf		
DecTr	fitctree	predict		
kNN	fitcknn	predict		
SVM	fitcsvm	predict		
LogReg	mnrfit	mnrval		
Neur	feedforwardnet; train	net		
Discr	fitcdiscr	predict		
Bayes	fitNaiveBayes	posterior		
HMM	hmmtrain	hmmdecode	kmeanlbg; disteusq	
ARIMA	vgxset; vgxvarx	NTA classifiers	aicbic	
RNN	layrecnet train	net		

Segment modelling and classification

With respect to segment-based approaches for the introduction of temporal information, the following methods are proposed: HMMs (Rabiner, 1989). This is a genuinely temporally-aware classifier which takes every sound frame xi ∈ ℝD and assigns it with a discrete label (Linde, Buzo & Gray, 1980; Brookes, 2006), thereby obtaining an observation Oi, which is an integer number ck (a code) in the range [0, C − 1]. The series of observations is assumed to be produced by an HMM made up of N connected states S, where the Sa state emits the observation code ck with an emitting probability Eak, and evolves to the state Sb with a transition probability Tab. For the recognition of isolated ‘words’ (anuran calls), with a distinct HMM designed for each class, a left–right model is the most appropriate, and the number of states should roughly correspond to the number of sounds (phonemes) within the call. However, the differences in error rate for values of N that are close to 5 are small. The structure and the value of N have been taken from Rabiner (1989) and they are depicted in Fig. 2. The E and T matrices are obtained for each class θ from their corresponding pattern frames Πθ using the forward–backward algorithm (Baum & Eagon, 1967). Once the HMMs are identified, the algorithm takes the series of observations for the full sound segment to be classified (and not only for a single frame), and estimates the probability of being produced by the HMM of each class. The full sound segment is labelled as belonging to the class with the highest probability. When a sound file has to be classified, three alternatives for the determination of the segment length have been explored: The full sound file (HMM-F).

A segment with the same length as the regions of interest (ROI) mean length (HMM–ROI). The ROIs are the segments of the sound patterns containing a valid sound (no silence or noise).

A segment defined by a SW of a certain length (HMM–SW).

Figure 2 Hidden Markov Model structure for each anuran call as proposed in Rabiner (1989).

This is the classifier recommended in the MPEG-7 standard. In this technique, the r most significant values have been used, where r is a parameter to be chosen from the experimentation. Additionally, the HMM classifiers tested in the paper use a 256-code (C = 256) quantization codebook.

RNNs (Parascandolo, Huttunen & Virtanen, 2016). The series of frame features xi is introduced into a neural network with H neurons in its hidden layer, which produces an intermediate output yi. The previous outputs yi−1 to yi−T are then introduced as new inputs of the network (Fig. 3). A value of H = 10 neurons in the hidden layer is used throughout the paper.

ARIMA models (Box, Jenkins & Reinsel, 2011). The series of frame features xi ∈ ℝD is considered the result of the vector ARIMA time series, VARIMA(a,d,b), defined as (8) xi(d)=C0+∑k=1aAkxi−k(d)+∑k=1bBkεi−k+εi,

where a is the order of the autoregressive model, d is the degree of differencing, and b is the order of the moving-average model. The coefficient matrices Ak and Bk have a D × D dimension, and the C0 vector, representing the time series mean, has D components. In this case, the number of features describing the sound segment is (a + b) × D2. For the sake of simplicity, the stationarity of time series (d = 0) is assumed. On the other hand, VARMA models can be approximated by equivalent VAR models (b = 0). In this case, the optimum value for the remaining order of the model (a) is obtained using the Akaike information criterion (AIC) (Akaike, 1974), and the Ak matrices are estimated using a maximum-likelihood technique (Hevia, 2008). Every sound segment, featured with a × D2 parameters, can now be labelled using NTA classifiers.

In order to determine the order of the model (a), first the optimum order for every k-th ROI pattern (in the training dataset) is computed using a weighted AR mean order a¯k, derived as (9) a¯k=∑i=1OMi·AICik∑i=1OMAICik,

where AICik is the AIC value for the k-th ROI pattern modelled as a VAR model of order i, and OM is the maximum VAR order considered (OM = 10 is used). The optimum value for the VAR order model is then determined by (10) a=1NROI∑iNROIa¯k,

where NROI is the number of ROI segment patterns.

Figure 3 Recurrent neural network structure.

Classification performance metrics

The definition of the proper performance indicators constitutes a key aspect in the evaluation of procedures, and it is difficult to overstate its importance (Sturm, 2014). In order to compare the results obtained for every classifier, several metrics for the performance of a classifier can be defined (Sokolova & Lapalme, 2009), all of which are based on the confusion matrix (Table 4).

Table 4 Confusion matrix.

	Classification class	
Classified as positive	Classified as negative	
Data class	Positive	TP (true positive)	FN (false negative)	
Negative	FP (false positive)	TN (true negative)	

The most relevant metrics and their definitions are shown in Table 5, where they are computed for each class that is considered ‘positive,’ as compared to the remaining classes, which are considered ‘negative.’ Additionally, an average value per class can be defined for each metric.

Table 5 Classification performance metrics based on the confusion matrix.

Metric	Formula	Evaluation focus	
Accuracy	ACC=TP+TNTP+TN+FP+FN	Overall effectiveness of a classifier	
Precision	PRC=TPTP+FP	Class agreement of the data labels with the positive labels given by the classifier	
Sensitivity	SNS=TPTP+FN	Effectiveness of a classifier to identify positive labels. Also called true positive rate (TPR)	
Specificity	SPC=TNTN+FP	How effectively a classifier identifies negative labels. Also called true negative rate (TNR)	
F1 score	F1=2PRC·SNSPRC+SNS	Combination of precision (PRC) and sensitivity (SNS) in a single metric	
Geometric mean	GM=SNS·SPC	Combination of sensitivity (SNS) and specificity (SPC) in a single metric	
Area under (ROC) curve	AUC=∫01SNS·dSPC	Combined metric based on the receiver operating characteristic (ROC) space (Powers, 2011)	

Since, in the dataset, the number of instances in every class remains imbalanced (see Table 1), the use of accuracy or precision as the main performance metric can imply a significant skew (Chawla, 2005). It is therefore preferred to use sensitivity and specificity since these remain unbiased metrics even when the classes are imbalanced (Gonzalez-Abril et al., 2014, 2017). Therefore, when a single metric is required for the comparison of classifier results (i.e. to identify ‘the best classifier’), the geometric mean or the area under curve (AUC) values are preferred since they combine, in a single metric, the sensitivity and the specificity which both present better behaviour in the presence of imbalanced classes. The AUC is more commonly employed and is the metric used for the selection of the best options and/or classifiers throughout the paper. When only one point is available in the receiver operating characteristic (ROC) space, the value of the AUC is computed as the arithmetic mean of sensitivity and specificity.

Confidence interval of the classification performance metrics

Once the classification performance metrics are obtained, it is good practice to estimate the confidence interval of their values. To undertake this task, a bootstrap analysis is performed (Efron & Tibshirani, 1994). Firstly consider the testing dataset 𝒯 containing S sounds. From this dataset, S samples are then taken with replacement and a new 𝒯1 dataset is obtained. Due to the replacement in the sampling process, certain sounds are not contained in 𝒯1, while others are repeated at least once. The classification metrics vector μ1 can now be computed for the 𝒯1 dataset.

This process is repeated Nb times (usually a large number), thereby obtaining datasets T1,T2⋯TNb and their corresponding metrics vectors μ1,μ2,⋯μNb. This set of metrics vectors is employed to estimate the pdf of the metrics vector f(μ) and other related statistics. This procedure is commonly employed to derive the confidence interval of the classification metrics. Therefore, considering the metric μk, which is the k-th metric in the μ vector, and its pdf fk(μk), the confidence interval of μk, for a given confidence level γ, is the interval between the values uk and vk such that Pr[uk ≤ μk ≤ vk] = γ. The value of uk can be estimated as the γ/2 percentile of μk, and the value vk as the 100 − (γ/2) percentile. Throughout this paper, bootstrap analysis with Nb = 1,000 and a confidence level of γ = 95% is used.

Bootstrap analysis can also be employed to estimate the probability that a certain metric outperforms another. For every 𝒯j dataset, the classification methods 1 and 2 are employed and their metric vectors μj1 and μj2 are computed. The difference between these metric vectors is then derived by δj = μj1 − μj2. The pdf of the differences vector f(δ) and the continuous density function (cdf), F(δ), can then be computed. Finally, considering the difference δk, which is the k-th metric in the δ vector, and its cdf Fk(δk), the probability of outperforming, ok, is the probability that δk > 0, that is, ok = Pr[δk > 0] = Fk(0).

Results

Instance selection vs. cross-validation

In Fig. 4, cross-validation and instance selection approaches are depicted for NTA classification based on 18 MPEG-7 features (NS-18). As can be observed, most of the algorithms present a significantly better performance when the patterns are chosen using the instance selection method, with an increase of more than seven points (in %) in the AUC metric of the centroid. Similar results are obtained for other temporally-aware and NTA classifiers for which instance selection has been employed.

Figure 4 Cross-validation and instance selection ROC analysis for non-temporally-aware classifiers based on 18 MPEG-7 features.

NTA classification for a varying number of features

The results obtained by the NTA classifiers based on 18 MPEG-7 features are compared using the ROC analysis, which is depicted in Fig. 5. The best result corresponds to the minimum distance classifier, with an AUC of 83.5%. This result is considered as the original baseline (denoted NTA-18) for future comparisons.

Figure 5 ROC analysis for non-temporally-aware classifiers based on 18 MPEG-7 features.

In order to prevent a high number of features entering the following temporally-aware algorithms, it could be convenient to reduce their number by selecting the r most relevant MPEG-7 features. To determine the value of r, the AUC of the validation dataset is used. In Fig. 6, the AUC values for three NTA classifiers are considered as a function of the number of features. The classifiers in the figure are those showing the best AUC performance for values of r: minimum distance, maximum likelihood, and decision tree. From this figure, it can be seen that using the 11 most relevant features (r = 11), the best AUC is obtained. On the other hand, if the computing effort is a major concern and therefore the number of features becomes an important issue, selecting the five most relevant features (r = 5) is a good balance between the AUC and the number of features. A further reduction would produce the steepest AUC decrease (below 75%, decreasing more than 10 points) which is confirmed below (joint optimization subsection). The following subsections derive the results for both the reduced (r = 5) and the optimum (r = 11) number of features.

Figure 6 AUC vs. the number of features for the three best non-temporally-aware classifiers.

Classification with a reduced number of features

NTA classification

For comparison purposes, the results obtained by the NTA classifiers based on the five most relevant MPEG-7 features are also compared using the ROC analysis, which is depicted in Fig. 7. The best result corresponds to the decision-tree classifier with an AUC of 80.7%. This result is considered as the reduced baseline (denoted NTA-5) for future comparisons.

Figure 7 ROC analysis for non-temporally-aware classifiers based on the five most relevant MPEG-7 features.

Determining the number of frames

In four of the proposed temporally-aware methods (SW, RSW, HMM–SW and RNN) several consecutive frames have to be considered. The first issue is therefore to determine the optimum number of frames (also called window size w). For this purpose, the AUC of the validation dataset is used, which is represented in Fig. 8 for several temporally-aware methods (using the best underlying NTA classifiers) as a function of the number of frames. In that figure, instead of the AUC absolute value, the increase in the AUC is depicted, compared to the w = 1 case. This graphical approach clearly shows the advantage of using temporally-aware classifiers. In all the methods, except in the RNN, only an odd number of frames have been considered because they are preferred in those algorithms.

Figure 8 AUC vs. the number of frames for several non-temporally-aware classifiers (five features).

From this figure, by using a window size between three and nine in the SW method, the AUC value can be enhanced by more than six points (in %). With these considerations, a seven-frame SW (w = 5) has been selected (its optimum value is denoted in the figure by a filled blue marker). This means a duration of 70 ms which roughly corresponds to seven opening periods of the anuran vocal cords. Similarly, the optimum values of the number of frames for the remaining methods are RSW: 11; HMM–SW: one; and RNN: 15.

LIQR classification

The first frame-based approach to temporally-aware classification is now considered: that of the construction of the LIQR features. The results corresponding to the ROC analysis are depicted in Fig. 9. The best result corresponds to the decision-tree classifier that has an AUC of 85.2%. For most of the classifiers, the LIQR approach attains slightly better results than does the equivalent NTA classifier: a mean enhancement of about five points (in %) is achieved in the AUC value compared to the reduced baseline (NTA-5).

Figure 9 ROC analysis for non-temporally-aware classifiers using LIQR feature construction.

SW classifiers

By considering the five most relevant features and a seven-frame window size, the SW method (SW7-5) is examined and its results compared through the ROC analysis, as presented in Fig. 10. The best result corresponds to the decision-tree classifier, with an AUC of 86.7%, which means an enhancement of about six points (in %) compared to the reduced baseline (NTA-5), and an enhancement of about three points (in %) compared to the original baseline (NTA-18).

Figure 10 ROC analysis for non-temporally-aware classifiers using the sliding window method.

The third frame-based approach to temporally-aware classification is now considered: the recurrent SW method. The results corresponding to the ROC analysis are depicted in Fig. 11, when five features (r = 5) and an 11-frame window size (w = 11) are considered (RSW11-5). The best result corresponds to the decision-tree classifier, which presents an AUC of 72.7%. For most of the classifiers, the recurrent SW approach obtains worse results than the equivalent NTA classifier, with a mean decrease of about 13 points (in %) in the AUC.

Figure 11 ROC analysis for non-temporally-aware classifiers using the recurrent sliding window method.

Segment-based classifiers

The HMM is the first segment-based approach to the introduction of the temporal information into the classification process. The HMM takes a sound segment and attempts to classify it as a whole, without any framing. The results corresponding to its ROC analysis are depicted in Fig. 12. In this figure, the five most relevant features (r = 5) are considered. The HMM over a segment defined by a SW (HMM–SW) of size w = 1, that is, over a single frame, obtains the best results among the HMM classifiers, with an AUC of 63.2% which, comparatively, is a poor result. Although the HMM is the classifier recommended in the MPEG-7 standard, it is clearly superseded by other NTA techniques.

Figure 12 ROC analysis for HMM and RNN classifiers.

The second segment-based approach to temporally-aware classification is now considered: the RNN. Using a hidden layer with H = 10 neurons, five features (r = 5), and 15 frames (w = 15), that is, a number of T = 14 previous intermediate outputs, an AUC of 61.0% has been obtained. This result is also depicted in Fig. 12.

Finally, the ARIMA segment-based approach is considered. As stated before, a vector AR (VAR) simplified model is considered, where the five most relevant features are used (r = 5). The first step is to determine the order of the VAR model (a) using the AIC criterion on the training dataset, as was described in the “Method” section. The results are depicted in Fig. 13, where the AIC values have been normalized to the [0,1] interval. A white point is drawn at every k-th row indicating the weighted AR mean order ak for the k-th ROI pattern. The optimum value for the VAR order model is represented in the figure with a vertical white line, and has the value a = 3.36. Its closest integer is used as the VAR order model, a = 3.

Figure 13 AIC values for ROI segment patterns.

Once the ARIMA models are determined, their parameters are classified using NTA classifiers and their performances are also compared using the ROC analysis, as illustrated in Fig. 14. The best result corresponds to the decision-tree classifier with an AUC of 62.0%.

Figure 14 ROC analysis for temporally-aware classifiers using ARIMA models.

Comparing classifiers

Hitherto, partial results have been presented for every temporally-aware method. In order to obtain an overall perspective, a comparison of the six different methods proposed for temporally-aware classifiers is presented in Fig. 15 and Table 6, where the NTA classifiers (original and reduced baselines) are also considered for reasons of contrast (best results are shown in bold).

Figure 15 AUC values for temporally-aware methods (five features).

Table 6 Summary of performance metrics (five features).

Method	Features	Frames	Best classifier	ACC	PRC	SNS	SPC	F1	GM	AUC	
NTA-18 (original baseline)	18	–	MinDis	84.12	55.37	76.95	90.03	64.40	83.23	83.49	
NTA-5 (reduced baseline)	5	–	DecTr	86.82	76.45	72.60	88.77	74.47	80.28	80.68	
LIQR	10 (2 × 5)	–	DecTr	91.88	79.25	77.49	92.94	78.36	84.86	85.22	
SW	35 (7 × 5)	7	DecTr	92.59	74.11	79.07	94.28	76.51	86.34	86.67	
RSW	55 (11 × 5)	11	DecTr	83.41	57.74	58.52	86.96	58.13	71.34	72.74	
HMM-F	5	–	–	75.41	44.64	40.62	82.78	42.53	57.99	61.70	
HMM–ROI	5	–	–	71.88	44.87	42.69	75.80	43.75	56.88	59.25	
HMM–SW	5	1	–	72.35	47.08	45.09	81.26	46.06	60.53	63.18	
RNN	75 (15 × 5)	15	–	66.59	47.81	40.01	81.91	43.58	57.27	60.98	
ARIMA	75 (3 × 52)	–	DecTr	80.94	38.75	38.47	85.50	38.61	57.35	61.98	
Note:

Best results are shown in bold.

Additionally, a ROC analysis has also been accomplished for every method and its results are depicted in Fig. 16. From these results, it can be observed that the best performance corresponds to the SW approach (with an underlying decision-tree classifier). It shows the best AUC metric with a value of 86.7%. The SW method also has the best values for almost every performance metric. The only exceptions are the precision and the F1 score (which depends on precision) which, although they present the highest values for the LIQR method, also present good figures for the SW method.

Figure 16 ROC analysis for temporally-aware methods.

Bootstrap analysis

Using bootstrap analysis on the testing dataset, the pdf of the classification performance metrics can be obtained. The results, focusing on the best temporally-aware method (SW) and considering the AUC pdfs for different window sizes, are shown as a colour map in Fig. 17. The colours represent the probability density for every AUC given a certain window size.

Figure 17 Colour map for the probability density function of the AUC vs. window size.

Results obtained using bootstrap analysis of the sliding window method.

Let us now centre on the SW method using the seven-frame optimum window size (as obtained using the validation dataset) and the reduced number of five features. This case, which is denoted as SW7-5, is now compared to the two NTA baselines: one with the original number of features (NTA-18), and the second with the reduced number of features (NTA-5). The pdfs for the AUC in these three cases are depicted in Fig. 18.

Figure 18 Probability density function of the AUC for the optimum sliding window case (with reduced number of features).

Comparison to the original and reduced baselines.

Not only can Bootstrap analysis offer the confidence interval for every classification performance metric, but it can also, even more importantly, show how much the optimum temporally-aware classification method (sliding window SW7-5) improves the results above the two NTA baselines (NTA-5 and NTA-18). The results for a 95% confidence level are shown in Table 7.

Table 7 Performance improvement of the sliding window method (five features, seven frames).

Performance improvement	ACC	PRC	SNS	SPC	F1	GM	AUC	
Baseline NTA-5	Mean	5.77	−2.33	6.50	5.51	2.06	6.08	6.01	
Conf. Int.	±2.70	±10.3	±12.3	±2.08	±9.94	±7.38	±6.72	
Baseline NTA-18	Mean	8.46	18.73	2.20	4.25	12.12	3.15	3.22	
Conf. Int.	±2.82	±6.94	±11.9	±1.93	±7.98	±7.05	±6.51	

The AUC improvement over the two mentioned baselines obtained via the SW method for various window sizes is depicted in Fig. 19. In this figure, the 95% confidence interval is also shown.

Figure 19 AUC improvement for the sliding window method with reduced number of features.

Comparison to the reduced (A) and original (B) baselines.

Classification with the optimum number of features

Separate optimization

It is now time to turn our attention to the cases when the number of features is not such an important issue and it is affordable to use the r = 11 most relevant features. This number was obtained through an optimization procedure presented above.

Now, again, the next issue is to run a second and separate optimization process to determine the optimum number of frames for the methods requiring such a parameter. For this purpose, the AUC on the validation dataset is used, which is represented in Fig. 20 for several temporally-aware methods (using the best underlying NTA classifiers) as a function of the number of frames. In this figure, instead of the AUC absolute value, the increase of the AUC compared to the w = 1 case is depicted.

Figure 20 AUC vs. the number of frames for several non-temporally-aware classifiers (11 features).

From this figure, by using a window size between three and nine in the SW method, the AUC value can be enhanced by about three points (in %). With these considerations, a three-frame SW (w = 3) has been selected (its optimum value, denoted in the figure by a filled blue marker). Similarly, the optimum number of frames for the remaining methods are RSW: three; HMM–SW: 11; and RNN: three.

Repeating the analysis of the various temporally-aware methods on the testing dataset, now using 11 features, the results obtained are presented in Fig. 21 and Table 8, where the NTA classifiers (original and optimum baselines) are also considered for reasons of contrast (best results are shown in bold).

Figure 21 AUC values for temporally-aware methods (11 features).

Table 8 Summary of performance metrics (11 features).

Method	Features	Frames	Best classifier	ACC	PRC	SNS	SPC	F1	GM	AUC	
NTA-18 (original baseline)	18	–	MinDis	84.12	55.37	76.95	90.03	64.40	83.23	83.49	
NTA-11 (reduced baseline)	11	–	MinDis	88.00	77.67	81.03	92.50	79.31	86.58	86.77	
LIQR	22 (2 × 11)	–	DecTr	89.77	75.89	74.59	91.49	75.23	82.61	83.04	
SW	33 (3 × 11)	3	MinDis	90.47	79.30	82.85	93.93	81.03	88.21	88.39	
RSW	33 (3 × 11)	3	MinDis	90.47	79.30	82.85	93.93	81.03	88.21	88.39	
HMM-F	11	–	–	78.24	21.77	50.42	85.20	48.13	65.54	67.81	
HMM–ROI	11	–	–	72.59	85.96	48.39	76.03	61.92	60.66	62.21	
HMM–SW	11	11	–	75.29	45.56	37.02	83.82	39.16	55.70	60.42	
RNN	33 (3 × 11)	3	–	71.06	48.69	47.44	84.56	48.06	63.34	66.00	
ARIMA	363 (3 × 112)	–	MinDis	89.29	48.03	47.88	90.81	47.96	65.94	69.34	
Note:

Best results are shown in bold.

Additionally, an ROC analysis has also been accomplished for every method and its results are depicted in Fig. 22. From these results, it can be observed that the best performance corresponds to the SW (and to the RSW) approach (with an underlying minimum distance). It shows the best AUC metric with a value of 88.4%. The SW (and RSW) method also has the best values for every performance metric. Although SW and RSW methods show exactly the same performance metrics on the testing dataset, the SW has been chosen as the best method for two reasons: it offers slightly better AUC on the validation dataset; and it provides better performance for non-optimum window sizes (see Fig. 20).

Figure 22 ROC analysis for temporally-aware methods (11 features).

Bootstrap analysis can now offer the confidence interval on how much the optimum temporally-aware classification method (sliding window SW3-11) improves the results above the two NTA baselines (NTA-11 and NTA-18). The results for a 95% confidence level are shown in Table 9.

Table 9 Performance improvement of the sliding window method (11 features, three frames).

Performance improvement	ACC	PRC	SNS	SPC	F1	GM	AUC	
Baseline NTA-5	Mean	2.47	1.64	1.85	1.43	1.74	1.66	1.64	
Conf. Int.	±2.76	±3.76	±11.4	±1.76	±6.63	±6.62	±6.20	
Baseline NTA-18	Mean	6.36	23.94	6.06	3.91	16.70	5.08	4.98	
Conf. Int.	±2.94	±5.28	±11.6	±1.85	±7.02	±6.71	±6.25	

Joint optimization

In the previous section the numbers of features and frames were separately optimized, that is, firstly the optimum number of features was determined and, subsequently, the optimum window size for that value was derived.

However, it is also possible to run a joint optimization process to simultaneously seek the optimum values for both parameters. By running this process on the validation set for the best temporal-aware method (SW), a set of AUC values for each pair of values of the parameters (number of features and window size) is obtained. The result is shown in Fig. 23 in the form of several lines (one per window size) that depict the increase of the AUC compared to the w = 1 case. In this figure, the maximum values (optimums in the number of features dimension) are represented by small filled circles. This figure also confirms that the selection of the five most relevant features (r = 5) provides a good balance between the AUC and the number of features (a result previously derived from Fig. 6).

Figure 23 Increasing the AUC values for the sliding window method with a varying number of features and window sizes.

An alternative way to represent the joint optimization process is to employ a bidimensional colour map, as in Fig. 24, which depicts the increase of AUC for every pair of values (number of features, window size). The optimums in the number of features dimension are marked with a black spot, while the optimums in the window size dimension are denoted by an empty circle. The overall optimum value, which is indicated with a cyan filled circle, is reached for eight features and a seven-frame window (SW7-8).

Figure 24 Colour map of the increase in the AUC values for the sliding window method.

In order to ascertain the impact of optimizing in each direction, Fig. 25 has been constructed. Given a certain number of features (x-coordinate for the corresponding blue point), the AUC can be optimized by changing the window size, given by a vertical movement in Fig. 24, and the maximum value is the y-coordinate for that blue point. Alternatively, given a certain window size (x-coordinate for the corresponding green point), the AUC can then be optimized through the selection of the proper number of features, given by a horizontal movement in Fig. 24, and the maximum value is the y-coordinate for that green point. It can be seen that, in almost every case, the optimization of the window size offers greater improvement than the optimization of the number of features.

Figure 25 Impact of optimizing the AUC by selecting the window size or the number of features.

In Fig. 23, the AUC is plotted as a function of the number of extracted (primary) features, which has been denoted as D. However, the SW method adds other C = w·D constructed (secondary or derived) features. Therefore, the total number of features defining the dimension of the space for classification purposes is D + C, and this is the value which has to be considered and kept as low as possible in order to reduce the computing requirements. For this reason, it is worth redrawing this figure in terms of the total number of features. The result is shown in Fig. 26. It can be seen that, if the total number of features is a concern, then the green line (corresponding to w = 3 frames) has a suboptimum (a secondary peak identified with an empty green circle) corresponding to five extracted features, that is, 15 total features. The corresponding classifier (SW3-5) should also be considered as a possible alternative.

Figure 26 Increase of the AUC values for the sliding window method as a function of the dimension of feature space (D + C).

Summary of results

Throughout the previous subsections, several classification methods have been identified. Firstly, there are three NTA classifiers using: the original number of features (NTA-18), the reduced or balanced number of features (NTA-5) and the optimum number of features (NTA-11). These three classifiers have been used as the baselines to determine the improvement achieved using other procedures. Later, when considering temporally-aware methods, the SW method has shown itself to be the most efficient. The determination of the window size in a separate optimization process identifies a classifier for a balanced number of (primary) features (SW7-5) and another classifier for the optimum number of features (SW3-11). On the other hand, the joint optimization of the number of features and frames leads to the detection of an optimum method (SW7-8) or of a classifier that balances the performance metric and the dimension of feature space (SW3-5). Table 10 summarizes these seven classification methods.

Table 10 Summary of the best classification methods.

Method	Best classifier	Features	Temporally aware	Optimization	Features concern	
NTA-18	MinDis	18	No	No	Original	
NTA-5	DecTr	5	No	No	Balanced	
NTA-11	MinDis	11	No	Only features	No	
SW7-5	DecTr	35 (7 × 5)	Yes	Separate	Balanced	
SW3-11	MinDis	33 (3 × 11)	Yes	Separate	No	
SW3-5	DecTr	15 (3 × 5)	Yes	Joint	Balanced	
SW7-8	DecTr	56 (7 × 8)	Yes	Joint	No	

Using bootstrap analysis, the pdf of each performance metric for each classification method can be estimated. The results regarding AUC are shown in Fig. 27 with the classification methods ordered in terms of the increasing number of total features. It can be seen that all the SW classifiers (except the SW3-5) obtain very similar results and improve the original baseline by about five points (NTA-18).

Figure 27 Probability density function of the AUC.

However, by considering another classification performance metrics, different results can be obtained. For instance, Fig. 28 depicts the comparison of the bootstrap analysis when the accuracy (ACC) is considered. Now all the SW classifiers clearly outperform the NTA methods. The best classifier (SW7-8), which was obtained by a joint optimization process, increases the original baseline (NTA-18) by more than 10 points.

Figure 28 Probability density function of the ACC.

In order to compare each classification method using various performance metrics, a box plot has been drawn (Fig. 29). For each metric and each method, four elements are drawn: a filled box from the 25% to 75% percentiles; an upper vertical line from the 75% percentile to the upper limit of the confidence interval; a lower vertical line from the 25% percentile to the lower limit of the confidence interval; and a horizontal black line corresponding to the median value.

Figure 29 Box plot for each performance metric.

This same information is also presented in Table 11 (best results are shown in bold). Using AUC as the single performance metric, the overall best classifier is the SW3-11 which outperforms the baseline by about five points (requiring 33 features instead of 18). However, the SW7-8 classifier (requiring 56 features) outperforms SW3-11 in terms of accuracy, precision, specificity and F1 score, and stands as the second best in terms of AUC. On the other hand, if the number of features is the greatest concern, then the NTA-11 classifier (requiring 11 features), still outperforms the original baseline with a much reduced number of features.

Table 11 Performance improvement (%) over the original baseline (NTA-18).

Method	Features	Statistic	ACC	PRC	SNS	SPC	F1	GM	AUC	
NTA-5	5	Mean	2.69	21.06	−4.31	−1.27	10.06	−2.94	−2.79	
NTA-11	11	Mean	3.89	22.30	4.21	2.48	14.96	3.42	3.34	
SW3-5	15 (3 × 5)	Mean	7.05	20.60	−2.13	2.44	10.95	−0.06	0.16	
SW3-11	33 (3 × 11)	Mean	6.36	23.94	6.06	3.91	16.70	5.08	4.98	
SW7-5	35 (7 × 5)	Mean	8.46	18.71	2.20	4.25	12.12	3.15	3.22	
SW7-8	56 (7 × 8)	Mean	9.29	28.55	3.83	4.32	17.87	4.07	4.07	
Note:

Best results are shown in bold.

By considering not only the mean value of the improvements but also their statistical distribution, the confidence interval for each metric and method can be derived. These results are shown in Table 12, where the probability that the chosen method outperforms the original baseline (NTA-18) is also presented (best results are shown in bold). It can be seen that for almost every metric, the selected method outperforms the original NTA classifier with a high probability.

Table 12 Performance improvement (%) over the original baseline (NTA-18) with confidence interval.

Method	Features	Statistic	ACC	PRC	SNS	SPC	F1	GM	AUC	
NTA-11	11	Mean	3.89	22.30	4.21	2.48	14.96	3.42	3.34	
Conf. Int.	±2.94	±5.12	±11.5	±1.90	±6.94	±6.72	±6.29	
Pr. Outperf.	99.42	99.83	77.52	99.38	99.82	85.22	86.23	
SW3-11	33 (3 × 11)	Mean	6.36	23.94	6.06	3.91	16.70	5.08	4.98	
Conf. Int.	±2.94	±5.28	±11.6	±1.85	±7.02	±6.71	±6.25	
Pr. Outperf.	100	99.97	86.08	100	99.91	93.81	94.56	
SW7-8	56 (7 × 8)	Mean	9.29	28.55	3.83	4.32	17.87	4.07	4.07	
Conf. Int.	±2.71	±10.3	±12.0	±1.91	±9.01	±7.07	±6.54	
Pr. Outperf.	100	100	74.67	99.99	99.86	88.02	89.51	
Note:

Best results are shown in bold.

Discussion

We show that instance selection for the classification of the sounds in the training dataset offers better results than do cross-validation techniques. This is consistent with several other studies that have shown that selective learning helps reduce the effect of the noise in the data (Raman & Ioerger, 2003; Olvera-López et al., 2010). As has been addressed in Borovicka et al. (2012), many instances in the training set may prove to be useless for classification purposes and they commonly do not improve the predictive performance of the model and may even degrade it. Despite the noise in the data, certain researchers (Blum & Langley, 1997) have proposed a further two reasons for instance selection. The first reason arises when the learning algorithm is computationally intensive; in this case, if sufficient training data is available, it makes sense to learn from only a limited number of examples for purposes of computational efficiency. Another reason arises when the cost of labelling is high (e.g., when labels must be obtained from experts). In our case, the identification of the first and final frames of the ROIs is a burdensome task which can be minimized by using fewer examples in the training dataset.

Furthermore, from the results, the decision-tree method appears as one of the best classifiers in many temporally-aware methods. This fact is consistent with other studies where non-speech sounds (Pavlopoulos, Stasis & Loukis, 2004), or more specifically, environmental sounds (Bravo, Berríos & Aide, 2017) are considered.

Additionally, the temporally-aware classifiers have revealed that they can outperform their NTA counterparts. Several authors (Dietrich, Palm & Schwenker, 2003; Salamon et al., 2016) have reached similar results in the field of bioacoustics and argue that the constructed features can better capture spectro-temporal shapes that are representative of the various sound classes.

The SW method attained the best results in our tests, which is also consistent with other works (Salamon & Bello, 2015) that shows that feature learning is more effective when the learning is performed jointly on groups of frames. In their study the authors have reported very similar results for various window sizes. Our study, however, which has comprised a larger set of values for window size, concludes that there is an optimum region for the optimum number of frames and that overly large values of this parameter can even degrade the classification performance. The results attained by the SW method even outperformed the HMM usually employed in speech recognition applications. This result is mainly due to the fact that the HMM is a classifier that uses sub-word features, which are not suitable for non-speech sound identification since environmental sounds lack the phonetic structure that speech possesses (Cowling & Sitte, 2003). It has been found that the optimum classifier (SW3-11) increases the AUC by approximately five points and obtains a noteworthy overall accuracy of 90.5% (six points higher than the baseline). Since the level of background noise in the recordings is high, this can be considered a remarkable result. In Salamon & Bello (2015), an increase of 1.5 points in the AUC and five points in accuracy were reported for an eight-frame window size.

The outperformance using these methods may only be moderate but it is reliably consistent. The probability that the selected temporally-aware methods improve their NTA counterparts is extremely high (more than 90% in most cases).

Conversely, the cost of more complex computing due to the higher number of features required in the optimum SW3-11 method has been considered in detail (Luque et al., 2017). Using 33 (3 × 11) features almost double the number of the original 18 parameters which affects processing efforts in three different aspects. The first issue involves the time required for the construction of the new features that, for the three-frames SW optimum method, is approximately 10 μs measured on a conventional desktop computer. This time is negligible compared to the classification time (detailed below) and to the frame length (10 ms, 1,000 times higher).

Additionally employing a greater number of features leads to higher processing requirements in the task of training classifiers. By doubling the number of features, the time needed to train classifiers is also approximately doubled, with values of 30 ms for the minimum distance and of 800 ms for the decision tree. Although these values are greater than the 10 ms window length they have a limited effect on the overall classification process because classifiers are trained off-line only once and, therefore, they do not affect real-time performances.

A third issue regarding processing efforts is the effect of employing a greater number of features on classification times. Using 33 features instead of the original 18 parameters approximately increases the classification time of a frame by 5% with absolute values of approximately 2 ms for the minimum distance and 0.8 ms for the decision tree. Hence, a very limited rise in the computing effort is demanded when the temporally-aware methods are applied.

Another issue to be considered is the ability of the proposed method to identify when within each audio recording the call is located. It could be thought that SW classifiers are going to blur the edges of audio events by using features obtained over a wider time span. However, these classifiers and certain other temporally-aware methods can still sharply identify the events. The SW method features a frame considering preceding and subsequent frames, but it still independently classifies every frame, thereby allowing the precise identification of calls as has been shown in several independent studies (Mesaros, Heittola & Virtanen, 2016; Stowell & Clayton, 2015; Foggia et al., 2015).

Conclusion

Changes in the sounds of anurans can be used as an indicator of climate change. Algorithms and tools for the automatic classification of the different classes of sounds could be developed for this purpose. In this paper, six different classification methods based on the data-mining domain have been proposed, which try to take advantage of the temporal behaviour of sound. The definition and comparison of this behaviour is undertaken using several approaches.

A detailed analysis of the classification errors shows that most errors occur when the recordings are very noisy. Additionally, other misclassifications appear when a recording, labelled as belonging to a certain class, is in fact made up of two or more overlapping sounds: one belonging to the true class and the others to a false class.

Firstly, it has been shown that instance selection for the determination of the sounds in the training dataset offers better results than do cross-validation techniques.

Additionally, the temporally-aware classifiers have revealed that they can obtain a better performance than their NTA counterparts. The SW method attained the best results in our tests, and even outperformed the HMM usually employed in speech recognition applications.

For classifiers based on a given number of features, the optimization of the window size can increase the AUC value by up to 12 points (in %), while the optimization of the number of features only leads to an AUC increase of fewer than three points.

If the number of total features is of no great concern, then the optimum classifier for our dataset is based on 11 original features and a window with three frames (SW3-11), which increases the AUC by about five points and obtains a noteworthy overall accuracy of 90.5%: a result even more significant when one considers the high level of background noise affecting the sounds under analysis.

On the other hand, if the number of features has to be minimized due to low computing capacity then the optimization of the number of features in NTA classifiers presents the best method, with an optimum for 11 features (NTA-11) thereby achieving an increase in the AUC of three points. If a further reduction in the number of features is required, a good compromise is found in the use of only five features (NTA-5) instead of the original 18, which reduces the number of parameters to less than one third while it reduces the AUC performance by only three points.

Supplemental Information

Supplemental Information 1 A1-0496 recording.

Click here for additional data file.

Supplemental Information 2 A1-0497 recording.

Click here for additional data file.

Supplemental Information 3 A1-049 8 recording.

Click here for additional data file.

Supplemental Information 4 A1-05 00 recording.

Click here for additional data file.

Supplemental Information 5 A1-05 01 recording.

Click here for additional data file.

Supplemental Information 6 A1-05 60 recording.

Click here for additional data file.

Supplemental Information 7 A1-05 69 recording.

Click here for additional data file.

Supplemental Information 8 A1-0 697 recording.

Click here for additional data file.

Supplemental Information 9 A1-0 698 recording.

Click here for additional data file.

Supplemental Information 10 A1-0 699 recording.

Click here for additional data file.

Supplemental Information 11 A1-0 701 recording.

Click here for additional data file.

Supplemental Information 12 A1-0 704 recording.

Click here for additional data file.

Supplemental Information 13 A1-0 707 recording.

Click here for additional data file.

Supplemental Information 14 A1-0 769 recording.

Click here for additional data file.

Supplemental Information 15 A1-0 770 recording.

Click here for additional data file.

Supplemental Information 16 A1-0 771 recording.

Click here for additional data file.

Supplemental Information 17 A1-0 773 recording.

Click here for additional data file.

Supplemental Information 18 A1-0 774 recording.

Click here for additional data file.

Supplemental Information 19 A1-0 775 recording.

Click here for additional data file.

Supplemental Information 20 A2-0 519 recording.

Click here for additional data file.

Supplemental Information 21 A2-0 558 recording.

Click here for additional data file.

Supplemental Information 22 A2-0 559 recording.

Click here for additional data file.

Supplemental Information 23 A3-0 503 recording.

Click here for additional data file.

Supplemental Information 24 A3-0 505 recording.

Click here for additional data file.

Supplemental Information 25 A3-0 700 recording.

Click here for additional data file.

Supplemental Information 26 A3-0 703 recording.

Click here for additional data file.

Supplemental Information 27 A3-0 705 recording.

Click here for additional data file.

Supplemental Information 28 A3-0 706 recording.

Click here for additional data file.

Supplemental Information 29 A3-0 711 recording.

Click here for additional data file.

Supplemental Information 30 A3-0 712 recording.

Click here for additional data file.

Supplemental Information 31 A3-0 762 recording.

Click here for additional data file.

Supplemental Information 32 A3-0 763 recording.

Click here for additional data file.

Supplemental Information 33 B1-1321 recording.

Click here for additional data file.

Supplemental Information 34 B1-1322 recording.

Click here for additional data file.

Supplemental Information 35 B1-1324 recording.

Click here for additional data file.

Supplemental Information 36 B1-1325 recording.

Click here for additional data file.

Supplemental Information 37 B1-1326 recording.

Click here for additional data file.

Supplemental Information 38 B1-1327 recording.

Click here for additional data file.

Supplemental Information 39 B1-1328 recording.

Click here for additional data file.

Supplemental Information 40 B1-1329 recording.

Click here for additional data file.

Supplemental Information 41 B1-1330 recording.

Click here for additional data file.

Supplemental Information 42 B1-1331 recording.

Click here for additional data file.

Supplemental Information 43 B1-1332 recording.

Click here for additional data file.

Supplemental Information 44 B1-1333 recording.

Click here for additional data file.

Supplemental Information 45 B1-1334 recording.

Click here for additional data file.

Supplemental Information 46 B1-1335 recording.

Click here for additional data file.

Supplemental Information 47 B1-1337 recording.

Click here for additional data file.

Supplemental Information 48 B1-1338 recording.

Click here for additional data file.

Supplemental Information 49 B1-1340 recording.

Click here for additional data file.

Supplemental Information 50 B1-1341 recording.

Click here for additional data file.

Supplemental Information 51 B1-1342 recording.

Click here for additional data file.

Supplemental Information 52 B1-1343 recording.

Click here for additional data file.

Supplemental Information 53 B1-1344 recording.

Click here for additional data file.

Supplemental Information 54 B1-1345 recording.

Click here for additional data file.

Supplemental Information 55 B1-1346 recording.

Click here for additional data file.

Supplemental Information 56 B1-1347 recording.

Click here for additional data file.

Supplemental Information 57 B1-1348 recording.

Click here for additional data file.

Supplemental Information 58 B1-1349 recording.

Click here for additional data file.

Supplemental Information 59 B1-1350 recording.

Click here for additional data file.

Supplemental Information 60 B1-1351 recording.

Click here for additional data file.

Supplemental Information 61 B1-1352 recording.

Click here for additional data file.

Supplemental Information 62 B2-1353 recording.

Click here for additional data file.

Supplemental Information 63 Matlab code: Classification with variable number of parameters using cross-validation.

Click here for additional data file.

Supplemental Information 64 Matlab code: Classification with variable number of parameters.

Click here for additional data file.

Supplemental Information 65 Matlab code: Classification with variable number of frames.

Click here for additional data file.

Supplemental Information 66 Matlab code: Instance selection vs. cross-validation.

Click here for additional data file.

Supplemental Information 67 Matlab code: Classification with HMM (Hidden Markov Models) and recursice neural networks.

Click here for additional data file.

The authors would like to thank Rafael Ignacio Marquez Martinez de Orense (Museo Nacional de Ciencias Naturales) and Juan Francisco Beltrán Gala (Faculty of Biology, University of Seville) for their collaboration and support.

Additional Information and Declarations

Competing Interests

Author Contributions

Data Availability

The authors declare that they have no competing interests.

Amalia Luque conceived and designed the experiments, performed the experiments, analyzed the data, contributed reagents/materials/analysis tools, prepared figures and/or tables, authored or reviewed drafts of the paper, approved the final draft.

Javier Romero-Lemos performed the experiments, analyzed the data, contributed reagents/materials/analysis tools, prepared figures and/or tables, authored or reviewed drafts of the paper, approved the final draft.

Alejandro Carrasco performed the experiments, contributed reagents/materials/analysis tools, authored or reviewed drafts of the paper, approved the final draft.

Luis Gonzalez-Abril contributed reagents/materials/analysis tools, authored or reviewed drafts of the paper, approved the final draft.

The following information was supplied regarding data availability:

The recordings of the anuran sounds are provided as Supplemental Files.

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
