# Peer review of "Temporally-aware algorithms for the classification of anuran sounds"

_PeerJ, doi:10.7717/peerj.4732_

## Round 0.1 · original submission · Major Revisions

The two referees, both with considerable expertise in the area of bioacoustics, see merit in the paper. I agree. However, there remains much room for improvement before the paper can be considered for final publication. Reviewer 1, in particular, provided a detailed list of suggestions that should prove very helpful (and necessary) for restructuring and improvement of presentation. As such, I would like to see a revised manuscript that provides a point-by-point response to the reviewers, along with an indication of where appropriate changes have been made in the revised text. (Further details for how to undertake a revision are provided in the linked checklists, please follow these instructions carefully.)

Reviewer 1 ·

Basic reporting

The title should be changed: "sequential" is a confusing word here, since in bioacoustics "sequence" usually means a sequence of sound events, NOT a sequence of acoustic frames as in this paper. (See for example Kershenbaum et al "Acoustic sequences in non-human animals: a tutorial review and prospectus".)

The intro should make clearer what the goal of classification is: to recognise individuals? to recognise species?

English language could be improved. Specific examples: l26 "Let us indicate that"; l62 l73 "Every frame is featured"; l368 "ROC analysis has also been accomplished". In all these the verbs are odd choices.

Experimental design

I have a concern about method. Some parameters are selected by inspecting the results: e.g. l268 "To determine the value of r, the F1 score is used." The authors do not state if this F1 score is from the training data, the testing data, or held-out validation data. It is unfortunately quite common for some people to use the test data as a guide for parameter selection, but this is bad practice since it leads to over-optimistic estimates of performance, due to "peeking" at the testing data groundtruth: the testing data groundtruth should never be used for any part of the training or tuning process. The authors do not state how their parameters are tuned, only that they are. If the testing data has been used, various of the results reported should be called into question.
This issue arises on l268, l290, l361

Validity of the findings

Important validity question discussed above re "Experimental design".

Additional comments

This paper is about automatic classification of anuran sound recordings. The authors use a single dataset of anuran recordings, and evaluate various classifiers as well as various types of feature representation, in particular whether to classify individual acoustic frames or entire temporal sequences of them. The results find many methods that don't work very well, and a couple of temporally-aware methods that slightly outperform the baseline.

The paper is broadly OK and falls within a long tradition of papers that evaluate various tweaks of classifiers to find which gives the best results on a particular dataset.

The set of methods analysed is a mixture of very common (stacked frames in sliding windows, HMM) and unusual (recurrent sliding windows, local IQR). A useful outcome of the paper is that some common methods do not perform well for these authors. Note that the two strongest temporal methods perform only slightly better than the baseline "non-sequential" methods via decision-tree. We are given no confidence intervals on the performance (in principle those could be estimated from bootstrap sampling the results) so we do not know if this slight improvement should be considered significant. However, if taken as a case study of a range of methods applied to this dataset, various of the results would be interesting to a reader with an interest in such data.

The discussion section of the paper should really engage more with this issue. Is a small, possibly non-significant boost in score really worth the cost of more complex computation? Note also that the "non-sequential" method offers an extra benefit, of identifying when within each audio recording the call is located.

The paper is limited only to this one dataset, which is not uncommon practice - however it does mean that the generality of its conclusions is an unanswered question. The dataset is 3 classes corresponding to 2 species, and 63 recordings overall, making quite a small dataset to draw conclusions from.

The comparison of crossvalidation testing versus "instance selection" is a strength of the paper.


l72: "C additional parameters are constructed" - what does this mean? Authors must clarify. "Constructed" here is ambiguous.

l78 "The frame features are compared to..." - how are they compared? (The paper later states that it's via various methods. Meaning should be clear at every point)

l106-119: this new feature, the local IQR of feature values, they call "temporal parameters". That is an empty an unhelpful label (even if a previous work used that label). More helpful to the reader would be "local IQR features".

l120: Authors might like to note that "Sliding windows" has also been called "frame stacking" or "shingling" in various other works.

l139 claims that the hidden Markov model "quantizes" every sound frame. This is mistake. It assigns a discrete label to each frame, but it does not "quantize" it - that's a related but different concept. Must fix.

l156: "the r most significant values have been used" - the most significant values of what? What does it mean? How does one judge significance? Must be rewritten.

l182 "maximum likelihood" - under what probability distribution? It seems later that the authors are using Gaussian distributions (with full covariance? I don't know) - this must be stated clearly or else the classification model is unspecified.

l185 "neural networks" - which neural network? There are so many neural network types that it's not really appropriate to say, after testing one specific neural net, in a non-leading edge software framework (matlab), one has tested "neural networks".

l197: "the r most significant features of each frame" - again, what is "significant"? And is the same r as mentioned on l156 or not? Not clear.

l200 "Feature selection procedures are employed" - which ones? Without this info the method section is incomplete. Must fix.

l210 "Accuracy: Overall effectiveness of a classifier" - this is not a definition. How is it calculated? Must rewrite.

l211 "Precision: Class agreement of..." - "class agreement" is not clear how to calculate.

l221 the ROC statistic - this is a rather unusual metric. Much more common is to use the AUC metric. Why is AUC not used? Yes it can be calculated from a single point in ROC space.

l258 this para about method should not be in the results section, it should be in the method section.

Fig 6: add more description in the caption. The reader should be able to understand the figure. What is "best". What is the "mean" over?

l297 the authors claim that a window size of "between 2 and 7" emerges as best. However Fig 9 shows not a very clear picture - it seems there's no consistent pattern emergin. Authors should be more honest about this.

Fig 12: make the HMM points stand out more clearly please. e.g. greay out the others

l350 avoid "p" for a variable name; readers may confuse with probability.

l359 most of this paragraph is method not results - move it to method.

Table 3: please apply bold to the strongest value(s) in each column.

l399 "The sliding window with an underlying decision tree attained the best results" -- not a true statement. Must be fixed. In Table 3, the strongest results came most often from "temporal parameters", sometimes from "sliding window" and sometimes N18.

Reviewer 2 ·

Basic reporting

This manuscript aims to test different methods to classify anuran vocalizations. Anurans are a widely threatened group and acoustic monitoring is an important tool to evaluate changes in their populations. Considering the large amounts of information produced by acoustic monitoring, effective classifiers and recognizers of calls are of great importance to manage the information obtained.
As a general commentary, I think that although the techniques seem to be well applied and the manuscript may provide valuable information, the focus on anuran calls and climate change are not well developed in the text. The authors should try to improve the background provided in the introduction. Although traditional sections are provided (i.e., Introduction, Material & Methods, Results, Discussion and Conclusions), the text included in each section was not in fully accordance with the corresponding section. For instance, the scientific names of anurans (not well written) and the length of acoustic recordings appears for the first time in the Results section. In addition, there are formulas in this section. I think that a large amount of text in the Results sections should be moved to Material & Methods. The Discussion and Conclusion sections need more work. Perhaps the authors would benefit by reading the paper from Xie et al 2017 (Ecological Indicators, 82: 13-22). Until these general issues are not taken into account, I cannot provide more specific recommendations.

Experimental design

no comment

Validity of the findings

no comment

---

## Round 0.2 · Minor Revisions

The revised manuscript has been seen by one of the two original reviewers. In general, they are satisfied with the changes made and believe the manuscript has been strengthened as a result. However, they make an excellent point regarding the 'lightweight' impression given by the Discussion. It really needs further attention, along the lines suggested by the referee, with more context and reference to the broader literature and how this work fits with this past work. I would therefore invite a second revision to address this criticism more satisfactorily, as well as to remove redundant display items.

Reviewer 1 ·

Basic reporting

Much clearer and more completely reported.

Experimental design

Concerns about design have been addressed satsfactorily.

Validity of the findings

Yes.

Additional comments

The authors have successfully addressed my two main concerns from the first review:
(1) the confusing "sequential" language has been clarified, including in the much better title;
(2) the setting of classifier hyper-parameters, via training/validation/testing data splits has been clearly stated as part of the method and is satisfactory.

They have also implemented bootstrap confidence interval calculation as suggested, and have made productive use of this.

The "Discussion" section is not really a discussion, it's really a repetition of the main results. This is a real weakness of the present paper: the authors should DISCUSS the results, i.e. consider their importance, their connection to other knowledge in the field, their implications.
In my first review I said: "The discussion section of the paper should really engage more with this issue. Is a small, possibly non-significant boost in score really worth the cost of more complex computation? Note also that the 'non-sequential' method offers an extra benefit, of identifying when within each audio recording the call is located." This is now partly addressed in the "Conclusions" section. But it should be in the "Discussion" section, and should be more discursive.
The authors could merge the Discussion and Conclusions sections into one perhaps. The current Discussion section adds very little; it could be deleted. Ideally more thoughtful discussion would be added.

Minor point: some of the figures duplicate each other. Fig 17 not needed, since Fig 18 presents the same information in a more readable fashion.

---

## Round 0.3 · Minor Revisions

Thank you for the further revisions. I am satisfied that the content of the MS is now sufficient to warrant publication in PeerJ. However, I would like you to submit a final, minor revision, that attends to the following edits:

L70: delete
L77-80: here and throughout, put all latin names in italics
L370: Here and elsewhere when citing AUC, use only 1 decimal place. Given the inherent variability in the classifier, anything further is false precision.
L593: change "The preceding results first show" to "We show"
L611: "temporally-aware" remove dash
L616: combine this paragraph with the next (starting L623)
L628/623: combine paragraphs
L636: change "On the other hand" to "Conversely"
L650: remove "Finally, " and start with "A third issue"

---

## Round 0.4 · accepted · Accept

Thank you for addressing the final editorial points. The MS is now acceptable for publication - well done.

#